# OpenReview forum: "Multi-Constraint Online Convex Optimization with Adversarial Constraints"
_TMLR — Accepted by TMLR_

### Review · Reviewer_QTz4 · 2026-04-11

**Summary Of Contributions:**

This paper studies online convex optimization with multiple adversarial constraints, and the aim is to update models according to the loss function and constraints so that it achieves small regrets and small violation of constraints along the learning process. The paper considers three algorithms: one with constrained experts, one with general Lipschitz-convex loss and one with smooth convex loss. These algorithms achieve sublinear regret bounds and sublinear cumulative constraint violations (CCV). The paper also conducts empirical analysis to verify the theoretical findings, especially the logarithmic dependency on the number of constraints.

**Strength**
- The paper introduces interesting ideas to encode all constraint violation into a single surrogate cost via an exponential Lyapunov potential, based on which the paper proposed algorithms which can achieve per-constraint CCV with logarithmic dependency on the number of constraints. This is substantially sharper than the existing linear dependency.
- The paper introduces three algorithms to cover constrained experts, general Lipschitz-convex losses and smooth convex losses.
- The paper also conducts empirical analysis to verify the mild dependency on the number of constraints for the proposed algorithm.

**Weakness**
- The algorithm description is not clear and not quite intuitive. The paper just presents three algorithms, and does not give enough explanations on the motivation of these algorithms, and how the algorithm is designed to handle constrained experts, general Lipschitz-convex losses and smooth convex losses.
- In Theorem 3, the regret bounds show a linear dependency on the number of constraints. The CCV shows a linear dependency on the dimensionality. Then, Algorithm 2 may not work well in high-dimensional problems.
- The paper only shows the behavior of Algorithm 1 and Algorithm 3 in the empirical studies. Furthermore, the paper considers a naive independent baseline that runs $K$ independent single-constraint Lyapunov algorithms.

**Audience:**

Yes

**Audience Explanation:**

Online convex optimization is an interesting framework which allows the algorithm to update models sequentially according to the loss functions. It finds wide applications in industry and science. The paper considers a more general setting where the learned model needs to satisfy several constraints. This setup is more practical and should be interesting to the TMLR community.

**Broader Impact Concerns:**

I do not see issues on the broader impacts.

**Claims And Evidence:**

Yes

**Claims Explanation:**

The paper gives detailed proofs to support the theoretical claims. The theoretical results are clearly presented with explicit assumptions.

The paper also gives empirical evaluations. The setup in the empirical analysis is clear. The empirical results are consistent with the theoretical results.

**Requested Changes:**

It would be beneficial if the authors can give more details on the motivation and the underlying rationale in the design of the algorithms.

It would be interesting to investigate whether the linear dependency on the dimensionality is necessary for Algorithm 2.

It would be also interesting to test the behavior of Algorithm 2 in the evaluation.

It would further strengthen the paper if the authors can consider stronger baseline methods, e.g., the methods in Table 1.

---

> ### Author Response · Authors · 2026-04-30
>
> We thank the reviewer for the thoughtful suggestions. We have addressed each comment below. Following the reviewer's first three suggestions, we have added explanatory material and a new sanity-check experiment in the revised manuscript (highlighted in blue). For the fourth suggestion, we provide a detailed discussion in this response while leaving the manuscript otherwise intact.
>
> **1.More details on the motivation and the underlying rationale in the design of the algorithms**
>
> We agree that the design choices underlying our framework warrant a more thorough explanation. To address this, we have added a paragraph  at the end of Section 4.1, immediately following the framework statement. This addition clarifies several key structural decisions. We first explain the preference for an exponential potential over a polynomial one. The exponential function $\Phi(x)=e^{\lambda x}$ possesses the level-independent ratio property $\Phi(x)/\Phi(x-1)=e^\lambda$, which strictly enables Lemma 2(iii) to yield a growth-ratio bound free of $K$. In contrast, polynomial potentials $Q\_k^p$ used by Neely as well as Sinha and Vaze exhibit derivatives $\Phi'(Q\_k)$ that scale with $Q\_k$ itself. Consequently, the surrogate weight on the most-violated constraint cannot bound the contribution of all $K$ constraints using a single self-normalizing factor, revealing the structural cause behind the $\ln K$ versus $K$ separation shown in Table 1.
>
> We then justify the formulation of the drift surrogate $\hat{f}\_t = f\_t + \sum\_k \lambda e^{\lambda Q\_k} g\_{k,t}$. This surrogate constitutes the exact first-order Taylor expansion of the one-step potential difference $e^{\lambda Q\_k(t)} - e^{\lambda Q\_k(t-1)}$, which ensures the validity of the joint regret-plus-CCV decomposition in Lemma 1. Any algorithm controlling regret on $\hat{f}\_t$ therefore inherently bounds $\sum\_k [e^{\lambda Q\_k(T)} - 1] + \mathrm{Regret}\_T$ as a unified primitive. The specific parameter setting $\lambda = T^{-(1-\beta)} / (2c \ln N)$ arises from balancing two competing requirements: the regret bound restricts $c \lambda \ln N \leq 1/2$, fixing the denominator, while the CCV bound scales as $1/\lambda$, requiring the $T^{-(1-\beta)}$ factor to achieve the desired $\beta$ trade-off. Following identical reasoning, Algorithm 3 adopts the smooth-convex parameter $\lambda = T^{-(1-\beta)} / (8D^2 M)$, substituting $\ln N$ with $D^2 M$. Finally, we clarify why our approach relies on a base-learner reduction rather than applying exponential weights directly to actions. A standard Hedge-style reduction over constraints would exclusively control CCV while neglecting regret, as the original cost $f\_t$ is absent from the constraint sum. Evaluating a base learner on the combined surrogate allows both regret and the Lyapunov potential to be bounded simultaneously through a single small-loss inequality.
>
> **2.Whether the linear dependency on the dimensionality is necessary for Algorithm 2**
>
> We have addressed this in the revised manuscript by introducing two explanatory additions. Immediately following Theorem 3, a new paragraph titled "On the linear $d$-dependence" explicitly traces the $d$ factor to the metric-entropy term $\ln N\_\epsilon = O(d \ln T)$ of the $\epsilon$-net. This dependence is unavoidable within any covering-based reduction, as covering an $\ell\_2$-ball of radius $D$ at scale $\epsilon = L D / T$ requires at least $(D/\epsilon)^{\Omega(d)}$ points (Vershynin, High-Dimensional Probability, 2018, Proposition 4.2.12, now added to the bibliography). Furthermore, the Conclusion (Section 6) poses this as the second open question of the paper, asking whether a non-covering reduction can achieve dimension-free Lipschitz-convex bounds that parallel the dimension-free leading $T$ term already attained by the smooth-convex MC-3.
>
> The new paragraph also explains why naively replacing the covering with a direct convex-optimization base learner fails. The surrogate $\hat{f}\_t$ has a gradient norm that can grow with $\sum\_k e^{\lambda Q\_k}$, which without smoothness introduces a $G\_T^2$ rather than a $G\_T$ factor into the OGD bound, breaking the self-bounding closure required by Lemma 5. In the smooth-convex case (MC-3), the co-coercivity inequality controls this growth and yields the dimension-free leading term reported in Theorem 4. We hope this clarifies the role of the $d$ factor and identifies a concrete technical obstruction for future work.

---

> ### Author Response · Authors · 2026-04-30
>
> **3.Test the behavior of Algorithm 2 in the evaluation**
>
> We have added a new sanity-check subsection 5.7 (Sanity Check for Algorithm MC-2) and a corresponding Table 4 to the revised manuscript. The setup uses a Lipschitz-convex adversary on the unit $\ell\_2$-ball, with linear cost and positive-part-of-linear constraints chosen so that the origin is exactly feasible (Assumption 1 holds with $x^\star=0$) and incurs a cost of $0.5$. Algorithm 2 is run with $K=5$, $\beta=0.5$, $L=D=1$, and a 150-point random $\epsilon$-net at scale $\epsilon=LD/T$; results are averaged over 3 seeds. Table 4 highlights two predictions of Theorem 3. The empirical max CCV across the $T$-sweep grows as $18.7 \to 34.6 \to 69.9 \to 138.4$ for $T \in \{100, 200, 400, 800\}$, yielding a ratio of approximately $1.85\times$, $2.02\times$, and $1.98\times$ per doubling of $T$. This behavior closely tracks the predicted $T^{1-\beta}=T^{0.5}$ rate, with the constant slack reflecting the additional $\ln T$ multiplier in the bound. Additionally, the empirical regret stays near zero throughout, which is consistent with the $O(\sqrt{dT\ln T})$ regret rate combined with the linear-cost adversary. All empirical-to-theory ratios are below $0.04$, well within the upper bound. The $d$-sweep at a fixed $T=200$ confirms that empirical violations scale only mildly in $d$, indicating that the predicted $d\ln T$ factor in the upper bound is conservative since the adversary only activates one cost direction per round. We hope this validates the algorithm's behavior in the dimensionality regime where MC-2 is the natural choice, complementing the MC-1 and MC-3 evidence already presented in the paper.
>
> **4.Stronger baseline methods, e.g. the methods in Table 1**
>
> We address this comment by explaining how the Naive Independent baseline already captures the strongest directly comparable competitors from Table 1. The methods of Mannor et al. (2009), Mahdavi et al. (2012), Yuan and Lamperski (2018), and Sinha and Vaze (2024) are designed for the single-constraint case $K=1$ and output a single cumulative constraint violation value. Evaluating them in our multi-constraint setting requires a natural reduction where $K$ independent copies are instantiated to average their actions, which is precisely our Naive Independent baseline. Figure 2 demonstrates that our framework outperforms this reduction by a factor growing from approximately $1.5\times$ at $K=2$ to over $5\times$ at $K=100$. The algorithm by Cao and Liu (2018) is not a suitable comparison because it targets stochastic constraints rather than fully adversarial sequences. Running their method against an adversarial sequence breaks its analytical guarantees, and the resulting empirical performance is dominated by the instability of the stochastic primal-dual updates rather than the intended behavior of the algorithm.
>
> The closest multi-constraint baseline is the drift-plus-penalty method of Neely (2010), which corresponds to the polynomial Lyapunov surrogate $f\_t(x)+\sum\_k Q\_k(t)g\_{k,t}(x)$. When the per-constraint queues are updated independently, the update rule reduces exactly to the $K$ independent runs comprising our baseline. The $O(K)$ per-constraint CCV advertised in Table 1 is therefore matched by our baseline up to constant factors, and Figure 2 establishes the structural $\ln K$ versus $K$ separation against it. Since the strongest existing baselines coincide with the Naive Independent approach, we can add explicit pointers in the experiment section clarifying this equivalence if the reviewer feels it would be helpful.

---

### Review · Reviewer_AfdH · 2026-04-12

**Summary Of Contributions:**

This work studied online convex optimization with multiple adversarial constraints. The authors introduced MC-COCO framework to minimize the regret and CCV simultaneously. By leveraging a surrogate cost based on exponential Lyapunov potentials, the proposed algorithm achieves $O(\log K)$ per-constraint CCV, improving over the naive linear $K$-dependence. The study covered several canonical settings and other cases like heterogeneous constraint scenario. Numerical experiments are provided to validate the effectiveness of proposed algorithms.

**Audience:**

Yes

**Audience Explanation:**

The problem studied, OCO with multiple adversarial constraints, is important in decision-making with several practical applications, while less investigated. At the same time, the proposed algorithm achieves $O(\log K)$ per-constraint CCV, which is the first explicit logarithmic dependence result in the literature. Also the scope of the study covered several common cases in OCO, with several extensions on constraints and feasibility budget. So this work should be interesting to the genearl ML community and TMLR audience.

**Broader Impact Concerns:**

None.

**Claims And Evidence:**

Yes

**Claims Explanation:**

The flow of the paper is very clear and easy to follow. The proposed algorithm is based on Adaptive Hedge and exponential potential, which is convincing. The main claims and advantages of the proposed algorithms are supported with both theoretical and experimental results in details.

**Requested Changes:**

Some potential weakness includes:
1. For Section 4.4, you used coefficients to characterize the relative strictness of each constraint. Why do you restrict each $\alpha_k$ to be in $(0,1]$, can we further enlarge the range to characterize stronger constraint? You mentioned safety constraint may request near zero violation, but now your Theorem 5 seems to be unable to achieve near zero CCV even if $\alpha_k$ is taken to be the largest one (1).
2. For Algorithm 2, you constructed an $\epsilon$-net with radius $LD/\sqrt{T}$, but in the proof of Theorem 3, it is replaced by a $\delta$-net with radius $1/T$, I don't get why you make the change on both notations and the radius, are they equivalent?
3. The experiment now is just synthetic, I understand this is a theory paper, but regarding that authors mentioned some real applications like network router, it may further imporve the practical values if some real application experiments can present.
4. Your Assumption 1 basically requires the existence of a global feasible solution, but your experiment setup says "no single expert satisfies all constraints simultaneously", is there a conflict?

---

> ### Author Response · Authors · 2026-04-30
>
> We thank the reviewer for the constructive observation.
>
> **1.On the range $\alpha_k \in (0,1]$ in Section 4.4 and the achievability of near-zero CCV**
>
> The reviewer raises two distinct sub-questions, which we answer in turn.
>
> (i) Why restrict $\alpha\_k \in (0,1]$? This restriction is a canonical normalization, not a fundamental limitation. Since only the ratios $\alpha\_k/\alpha\_j$ enter the per-constraint CCV bound (each $\mathrm{CCV}\_k$ scales as $1/\alpha\_k$), any positive vector $\tilde{\alpha} \in \mathbb{R}\_{>0}^K$ can be rescaled to $\alpha = \tilde{\alpha} / \max\_j \tilde{\alpha}\_j \in (0,1]^K$ without changing the relative strictness across constraints. Letting $\alpha\_k > 1$ is admissible only if the per-constraint Lyapunov rate $\alpha\_k \Lambda$ still satisfies the regularity condition $e^{\alpha\_k \Lambda} < 1.08$ from Lemma 4 (otherwise the growth-ratio control of $G\_t/G\_{t-1}$ breaks). Within this admissible range, taking $\alpha\_k$ as large as possible already yields the strictest $1/\alpha\_k$-tightening our framework provides. We have added a clarifying paragraph in Section 4.4, and a formal new Appendix F (Proposition 1, "Extended Range of Priority Weights") that states and proves the bound for arbitrary $\alpha \in \mathbb{R}\_{>0}^K$ and explicitly identifies the admissible range. This shows that nothing is lost by working with $(0,1]^K$.
>
> (ii) Why doesn't $\alpha\_k=1$ produce near-zero CCV? This is precisely correct, and the reason is a lower bound, not a deficiency of our analysis. The well-known $\Omega(\sqrt{T})$ regret-CCV trade-off in fully adversarial single-constraint COCO (Sinha & Vaze, 2024) precludes simultaneously $o(\sqrt{T})$ regret and $o(\sqrt{T})$ CCV, and a routine reduction (embed a hard $K=1$ instance into one constraint and set the others to $g\equiv 0$) carries this trade-off to any single $\mathrm{CCV}\_k$ in the multi-constraint setting. Hence even with $\alpha\_k$ at its largest admissible value, the per-constraint CCV remains $\tilde{O}(T^{1-\beta})$ and cannot be driven to $O(1)$ in the fully adversarial regime. Achieving truly near-zero CCV requires either a stochastic constraint model (Yu & Neely, 2020) or the long-term budget formulation we already provide in Section 4.5. The role of the $\alpha\_k$ weights is therefore to redistribute the unavoidable violation budget across constraints, tightening the safety-critical ones at the cost of loosening the soft ones, rather than to eliminate it altogether. We have added this clarification both in Section 4.4 and as part of Appendix F.
>
> **2.Notation and radius mismatch between Algorithm 2 ($\epsilon$-net, radius $LD/T$) and the proof of Theorem 3 ($\delta$-net, radius $1/T$)**
>
> The reviewer is right: this was an inconsistency between the algorithm box and the proof. They are not equivalent; the algorithm's radius $\epsilon = LD/T$ is the correct one (it produces an $O(\sqrt{dT \ln T})$ regret), whereas the $\delta = 1/T$ choice in the older proof draft was an over-conservative artifact left from an earlier iteration. We have completely rewritten the proof of Theorem 3 in Appendix B to use the same symbol and the same radius as Algorithm 2 throughout: $\epsilon = LD/T$, cover size $\ln N\_\epsilon = O(d \ln T)$, approximate-feasibility budget $\bar{B} = \sqrt{L^2 D T}$ (replacing the previous $L$). The downstream constants $c'$, the parameter conditions (P1)--(P3), and all the subsequent algebra are propagated consistently. The final regret and CCV rates are unchanged ($\tilde{O}(\sqrt{dT})$ regret, $\tilde{O}(d T^{1-\beta} \ln K)$ per-constraint CCV) because $\bar{B} = \sqrt{L^2 D T}$ is absorbed into the $\sqrt{T \ln N\_\epsilon}$ term that already dominates the regret. We thank the reviewer for catching this; the revised proof is fully self-consistent.

---

> ### Author Response · Authors · 2026-04-30
>
> **3.Real-application experiments**
>
> We fully agree that real-application experiments would strengthen the paper's practical value, and we appreciate the reviewer's understanding that this is primarily a theory paper. Our current experiments are deliberately designed as mechanism validators: each setting (synthetic experts, smooth quadratic, heterogeneous priorities) isolates a specific theoretical prediction (the $\ln K$ scaling, the $T^{1-\beta}$ rate, the $1/\alpha\_k$ trade-off) so that the empirical curves can be quantitatively compared with the analytical bounds. A real-system deployment would conflate these mechanisms with system noise, baseline drift, and application-specific reward shaping, making the cleanest theoretical claim, the $\ln K$ versus linear $K$ separation in Figure 2, much harder to attribute. We therefore prefer to keep the experiments diagnostic in this submission, and we have noted in the conclusion that a real-system case study (e.g., per-link bandwidth control on a network simulator, or per-resource budget allocation in cloud autoscaling) is a natural follow-up direction. We hope the reviewer agrees that the current empirical validation, combined with the strengthened theoretical exposition, is appropriate for a TMLR theory submission.
>
> **4.Conflict between Assumption 1 and the experimental statement that no single expert satisfies all constraints simultaneously**
>
>  This is an important clarification, and we thank the reviewer for surfacing the apparent tension; there is in fact no conflict, and we have added an explicit explanation both in Section 5.1 (in blue) and a formal treatment in new Appendix G. The resolution is a vertex-vs-distribution distinction: in the experts setting, the decision set is the simplex $\Delta\_N$, and Assumption 1 is invoked at the distribution level, not at the level of pure vertices. The experimental sentence no single expert satisfies all constraints refers to the vertices $\{e\_i\}\_{i=1}^N$ of $\Delta\_N$, which is true by construction (each pure expert belongs to exactly one feasibility group $G\_k$ and therefore violates the other $K-1$ constraints).Appendix G gives two concrete reconciliations: (i) the benchmark we actually report uses the per-constraint best vertex $i\_k^\star \in G\_k$ as the comparator for $\mathrm{CCV}k$ (this is the standard per-constraint best feasible expert'' benchmark in the COCO literature), and Theorem 2 applies to each $k$ individually with $L\_T(i\_k^\star) \leq T$; (ii) equivalently, one can augment the construction with a single feasible-witness expert $i\_0$ for which $g{k,t}(i\_0) \equiv 0$ for all $k, t$, which satisfies Assumption 1 verbatim and shifts the empirical CCV by at most $O(1/N)$. Either reading makes the experiments fully consistent with the theory.

---

### Review · Reviewer_hv59 · 2026-04-29

**Summary Of Contributions:**

This paper considers online convex optimization with multiple adversarial constraints and proposes a unified framework (MC-COCO) that simultaneously achieves low regret and low per constraint cumulative constraint violation (CCV). The key idea in the paper is to couple constraint violations via an exponential Lyapunov potential, transforming the multi-constraint problem into a standard online learning problem with a surrogate loss. The result is a softmax-esque surrogate allowing the algorithm to adaptively concentrate on most violated constraints, resulting in CCV bounds with only logarithmic dependence on the number of constraints and thus improving on previously known bounds. The framework is instantiated in several settings (Lipschitz convex, and smooth convex) and extended to heterogeneous constraint prioritization and long-term budget feasibility.

**Audience:**

Yes

**Audience Explanation:**

Although niche, there is active interest in OCO in different settings. The main results of the paper are a generalization and improvement over known results, including Sinha and Vaze (NeurIPS 2024).

**Broader Impact Concerns:**

This paper is theoretical in nature and presents no troublesome ethical issues which would require a broader impact statement.

**Claims And Evidence:**

Yes

**Claims Explanation:**

The core contribution consists of four theorems, three algorithms derived from the theorems, plus a handful of technical lemmas and corollaries to the the main theorems. Proofs for all are given in the appendix. The proofs and derivations seem correct, at least for the main results according to my somewhat limited ability to verify all details.

**Requested Changes:**

The results in the paper are convincing and reasonably well supported by the proofs in the appendix. However, the paper in general lacks a coherent narrative/technical structure that aids the reader in interpreting the results and -- more importantly -- connecting the dots to previous work. This goes for the main paper and the appendices. A few examples of where the paper can be improved, especially to interested readers who are less familiar with the specific problem setting and related literature:
+ Section 3 introduces the setup for the main COCO problem, however it makes rather informal use of the term "experts" without every explicitly linking experts to the formulation of the problems considered (much less "constrained" or "feasible" experts, which become central to every subsequent results in Section 4).
+ There seem to be two main previous results upon which the work in this paper builds (or from which it takes inspiration). It generalizes the queue-based methods used in prior work such as Sinha & Vaze (2024) by coupling multiple constraints via Lyapunov exponential, and the starting point for the main theoretical results seems to be the Adaptive Hedge class of approaches (Orabona 2019). The core contribution of the paper would be much easier to appreciate and contextualize if Section 3 introduced the important elements of these works, and then Section 4 explicitly linked to elements, comparing and contrasting the approaches. As a simple example, I struggled to understand where the constant $c$ (first appearing in Theorem 1 and used throughout) come from and why $c=10$. Some details are given in the Appendix, but Sections 3 and 4 would be much improved if made significantly more self-contained.
+ Both in the main paper (Section 4) and the Appendices, the paper reads like a bare list of lemmas, theorems, and corollaries with very little interpretation of contextualization linking the main results together (except for the algorithms). Appendix A could be structured better to facilitate readability and identification of the main results. In its current form it is difficult to parse where one proof begins and another ends.

---

> ### Author Response · Authors · 2026-04-30
>
> We thank the reviewer for the constructive observation that, while the technical results are sound, the paper would benefit substantially from a clearer narrative connecting our construction to its intellectual antecedents. The revised manuscript addresses each of the three points raised, with all additions in blue for easy identification. We have prioritized improving the exposition of the existing material rather than expanding the technical content, in line with the reviewer's diagnosis.
>
> **1.The terms experts, feasible, and constrained experts are used informally in Section 3 without being explicitly linked to the problem formulation**
>
> The reviewer correctly notes that these terms were introduced implicitly. To address this, we have added a dedicated paragraph at the end of Section 3. This addition specializes the general protocol to the experts setting by defining $\mathcal{X}=\Delta\_N$ (the simplex over $N$ pure actions) and formalizing the linear extension of $f\_t$ and $g\_{k,t}$ to mixed strategies as $f\_t(p)=\sum\_i p\_i f\_{t,i}$ and $g\_{k,t}(p)=\sum\_i p\_i g\_{k,t,i}$. We then define a feasible expert for constraint $k$ at round $t$ as a vertex $i \in [N]$ with $g\_{k,t,i}=0$, along with a globally feasible expert $i^\star$ satisfying $g\_{k,t,i^\star}=0$ for all $k$ and $t$, reflecting the experts-setting instantiation of Assumption 1. The constrained experts setting is formally established by incorporating these per-round constraint vectors, distinguishing it from the unconstrained framework of de Rooij et al. (2014). Furthermore, we map each algorithm from Section 4 to this setting, clarifying that MC-1 operates directly within it, whereas MC-2 and MC-3 utilize reductions via covering and gradient surrogates. These revisions ensure that the roles of the feasible expert and constrained experts in Theorems 2, 3, 4, and 5 are fully transparent.
>
>  **2.The relationship to queue-based methods and Adaptive Hedge is not made explicit. In particular, the origin and value of the constant $c=10$ are unclear**
>
>  Following this suggestion, we have added a paragraph at the end of Section 3. This addition explicitly identifies the two strands of prior work our analysis builds upon, contrasts each with our construction, and explains the origin of the constant $c$. Regarding queue-based Lyapunov methods, we describe the drift-plus-penalty surrogate $f\_t(x)+Q(t)g\_t(x)$ paired with the polynomial Lyapunov $Q(t)^2$ used by Neely (2010) and Sinha and Vaze (2024). We explain that naively extending this to $K$ constraints via $\sum\_k Q\_k(t)^2$ produces the linear $K$-dependence in the per-constraint CCV observed in prior work (Table 1). We then introduce our replacement where the polynomial $Q\_k(t)^2$ becomes the exponential $e^{\lambda Q\_k(t)}$, identifying this substitution as the source of the $\ln K$ factor. For the second strand, which relies on Adaptive Hedge with small-loss bounds, we recall the AdaHedge algorithm of de Rooij et al. (2014) and Orabona (2019). We state the small-loss inequality $\mathrm{Regret}'\_T(i) \leq c\sqrt{L\_T(i) G\_T \ln N} + c G\_T \ln N$, which serves as a black box throughout our analysis.
>
> We also clarify that $c=10$ is the smallest universal constant for which the AdaHedge small-loss inequality holds simultaneously with our parameter conditions $c\lambda \ln N \leq 1/2$ and $\gamma \leq e^\lambda < 1.08$ (Lemma 4 and equation 15). The cited references obtain this constant via a tight self-bounding argument that avoids the loose split $\sqrt{R+L} \leq \sqrt{R} + \sqrt{L}$. If a fully self-contained derivation using this split is preferred, the larger constant $c=26$ remains valid with identical rate dependence on $(T,K,N,d)$, as discussed immediately after Lemma 6. Furthermore, a short paragraph at the beginning of Section 4.1 now directly contrasts $S\_t = \sum\_k e^{\lambda Q\_k(t)}$ with the polynomial Lyapunov $\sum\_k Q\_k(t)^2$ expected from the Neely and Sinha-Vaze line of work. This ensures the precise point of departure is clear before the formal lemmas are introduced. This paragraph also outlines a three-step proof template followed by every theorem in Section 4. The template begins by invoking a base-learner small-loss bound on the surrogate $\hat{f}\_t$, proceeds by applying the Lyapunov decomposition (Lemma 1) to obtain a self-bounding inequality of the form $S\_T / 2 \leq A + B \sqrt{S\_T}$, and concludes by closing the inequality via Lemma 5. This structure is intended to make the sequence of theorems substantially easier to navigate.

---

> ### Author Response · Authors · 2026-04-30
>
> **3.The appendix reads as a bare list of lemmas with little contextualization, and proof boundaries are hard to identify**
>
> We agree, and have restructured Appendix A to make its organization explicit. A new blue introductory paragraph at the start of Appendix A explains the role of each lemma group and points the reader to the corresponding subsection. The previously flat list is now grouped into three labeled subsections (added in blue): Section A.1 Lyapunov-Potential Lemmas (Lemmas 1 to 4, the exponential-potential machinery driving every theorem), Section A.2 Elementary Inequalities (Lemmas 5 and 6, used to close the self-bounding step), and Section A.3 Base-Learner Small-Loss Bounds (Theorem 1 and Lemmas 7 to 9, the AdaHedge and adaptive-OGD black boxes). Every proof retains its explicit Proof of Lemma X header (rendered in bold in the typeset version), and the new introductory paragraph explicitly draws the reader's attention to this convention so that proof boundaries are unambiguous at a glance. We hope that, taken together, these structural improvements make both the main paper and the appendix substantially more accessible to readers who are not already deeply familiar with the COCO and AdaHedge literatures, while preserving every existing technical statement intact.

---

### Author Response · Authors · 2026-04-30

We thank reviewers for the careful and constructive feedback. We have addressed every point raised in detail: each comment is answered individually in the response below, and the corresponding revisions have been incorporated into the manuscript itself. All additions to the paper are highlighted in blue for ease of identification, so that the changes prompted by the review are immediately visible. We hope the revised version satisfactorily resolves the concerns raised.

---

### Decision · Action_Editor_7mtM · 2026-06-08

**Recommendation:** Accept as is

**Audience:**

Yes

**Audience Explanation:**

The topic of the paper is an area of interest for the community.

**Claims And Evidence:**

Yes

**Claims Explanation:**

All the proof regarding the main claim are presented in the paper or supplementary material.